# OpenReview forum: "Unveiling the Invisible: Detection and Evaluation of Prototype Pollution Gadgets with Dynamic Taint Analysis"
_ACM.org/TheWebConf/2024/Conference — TheWebConf24 Oral_

### Official Review · Reviewer_cdTD · 2023-10-25

**Novelty:** 5
**Technical Quality:** 5

**Review:**

## Strength

+ The research problem, i.e., prototype pollution and its gadget detection, is important
+ The findings are interesting, resulting in a CVE with high severity

## Weakness

- The technical novelty is relatively low (mostly the detection capability comes from Silent Spring, another research paper)
- The technical contribution of the visualization is unclear
- No evaluation of false positives and negatives of the filtering

## Detailed Comments

Overall, this is an interesting paper that targets an important problem of gadget detection in Node.js applications.  I like the approach in general and think that the paper should have a place in the WebConf.  The paper also results in a high-impact vulnerability assigned with CVE identifiers.  However, at the same time, I also have the following concerns of the paper and hope that they can be addressed:

(1) The paper's research novelty is relatively low.  The research contribution, particularly the detection capability in gadget chain discovery, mostly comes from prior work, i.e., Silent Spring in USENIX 2023.   The proposed work is mostly a driver of the prior work, i.e., the USENIX's paper.  This is fine; then, the question is whether there exists other similar drivers that crawls NPM and applies vulnerability detection works.  This remains unclear to me after reading the paper.

(2) The paper mentions a UI component, which visualizes code flows with an IDE, thus facilitating the subsequent manual analysis for building proof-of-concept exploits.  The research contribution of this component is unclear.  Particularly, it remains unclear how easy it is to use this IDE for a regular user (besides the authors).  I think if we have this contribution claimed, the authors should conduct a user study to show that normal users are easy to use the IDE (compared with someone who does not have access to the IDE).  Then, the authors should report the user study results in the paper.

(3) The paper does not report false positive and negative numbers.  I understand that after manual verification, there is no false positive.  However, the paper should report false positives before manual verifications.  Also, the paper should consider an existing dataset (e.g., those that are manually verified) and report false negative numbers.  Those numbers will be helpful for the community to understand the effectiveness of the proposed tool.

**Questions:**

(1) Can you describe your technical novelty beyond Silent Spring?

(2) What is your research contribution to the proposed UI component?

(3) What are the FPs and FNs?

**Reviewer Confidence:**

4: The reviewer is certain that the evaluation is correct and very familiar with the relevant literature

**Scope:**

4: The work is relevant to the Web and to the track, and is of broad interest to the community

---

### Official Review · Reviewer_E7Fg · 2023-11-15

**Novelty:** 5
**Technical Quality:** 6

**Review:**

I very much enjoyed reading the work and do believe that studying prototype pollution gadgets in isolation is valuable to our community. Uncovering and remediating gadgets reduces the impact any potential prototype polluting vuln will have on the security of server-side applications.

Dasty is a nice tool to track data flows in Node packages with test suites and iterates on concepts established by prior research applying them to a new domain and uncovering exploitable gadgets in 49 real world packages. I appreciate the authors efforts in making Dasty available to the community upon publication.

With the information in the paper it is hard to quantify the impact of those vulnerabilities, I do understand that they are the most dependent-upon packages, yet, I believe the results could reap more impact given:
- context on the popularity of the packages (i.e., the dataset just says "most depended upon", but what does that mean for the packages that ended up being vulnerable)
- some in-depth analysis of the vulnerable data flows, e.g., complexity of flows, preconditions
- Likelyhood of being used in a context where prototype polluting vulns could be present, e.g., if a package is being mostly used in server-side CLI tools, probability of a pollutable prototype might be lower compared to the Kibana's of the world.

Another avenue to boost the impact of Dasty could be the addition of automatic exploit generation techniques similar to [43]. Having more information around the actual exploitable flows would allow future researchers to gauge feasibility and effectiveness of any automated exploit generation techniques. It would also allows us to scale the analysis to a bigger corpus of packages.

I would have liked to read more about the IDE visualization, but I assume that this fell victim to the page limit.

Regardless, I believe this is an interesting and impactful study.

Pros:
- First large-scale study of prototype pollution gadgets in the npm ecosystem
- Uncovered 49 actually exploitable gadgets in real-world packages
- Taint-analysis framework based on instrumentation to be made available to the community

Cons:
- Evaluation of uncovered vulnerabilities is sparse, making it hard to assess the impact of the found flaws
- Semi-automated approach with manual exploit generation and only 1/5 flows being exploitable, scaling this to the full NPM ecosystem would thus be hard

Nitpicks:
- You claim that related research does only focus on the pollution part, which is not the case for [19]. I do believe the domain is sufficiently different and the techniques of [19] are not immediately applicable to the node.js ecosystem, yet, I would tone down the claim a bit, as [19] is also concerned with gadgets and end-to-end exploits.
- Section 3.2-Execution sounds like the same strategy involved in [43]
- the "second" thread model does not seem to be in use, unless, if it implicitly references the end-to-end exploit of Kibana, I would consider dropping it to make the attacker model of the main parts of the work more crisp.

**Questions:**

- You claim that the analysis without force execution is not intrusive, however, changing out a native type with a proxy object will have issues that alter the control flow (as you also write in the paper). How big is the impact of this on the data set? How many crashes do you see that can be traced back to the proxified tainted values/the instrumentation?
- You exclude packages from your analysis that do not use Node.js APIs, how do you perform this analysis? Does this only concern the "main" package installed, or do you also check all of the dependencies's source code?

**Reviewer Confidence:**

4: The reviewer is certain that the evaluation is correct and very familiar with the relevant literature

**Scope:**

4: The work is relevant to the Web and to the track, and is of broad interest to the community

---

### Official Review · Reviewer_1mBU · 2023-11-22

**Novelty:** 6
**Technical Quality:** 7

**Review:**

## Summary
This work spans over the identification of gadgets that can be used in prototype pollution attacks exploiting Node.js applications. The authors perform dynamic taint analysis by extracting application entry points via tests provided in the application or developer provided test cases and identify their gadget candidates. Lastly, they combine their tool Dasty with existing prototype pollution identification tools to  discover new vulnerabilities and demonstrate their exploitability.


## Comments for the authors
I like the direction of this work and the way the paper is written which makes it easy to follow. I will share my thoughts and comments below:

### Comparison with related work
The authors report that they use Silent Spring [1] along with Dasty to identify the Kibana vulnerability. Given that Silent Spring also provides a hybrid gadget detection module, the authors should compare the performance and discuss the pros and cons of each approach.

### Removal of gadgets as a defensive mechanism
The authors compare the defense against prototype pollution to ROP exploits and argue that the solution is to remove such gadgets. I would argue that this solution is effective only if all if not majority of gadgets can be removed and merely reducing the number of gadgets would not prevent the attackers from modifying their exploit to rely on existing gadgets.

### Tests for Sinks
Dasty requires manual tests to exercise the code of the applications under test to allow the dynamic taint analysis to identify the gadgets. Given that this step seems integral for the proper functionality of Dasty, I would like to see more detail about the potential effects of such tests on the performance of Dasty. Is writing such a test a trivial task or does it require in depth knowledge of the code structure in the target npm packages? Would Dasty have identified more gadgets if the authors had access to better test cases compared to the existing unit tests that were extracted from the source code of packages under analysis?

### Default value and type extraction
In section 4 under Execution, the authors mention that their tool tries to infer the expected type and values based on the operations. What is the success ratio of this approach along with other orthogonal type extraction efforts and would the results be hindered if the type information cannot reliably be extracted?


### Suggested defense
Given that the mere presence of a gadget does not constitute a vulnerability, I find it important to discuss the suggested remedy from the perspective of the authors. What should the application developers do in the cases where a gadget is identified?

### Questions from the authors
Would forced execution explore branches where their conditions cannot feasibly be satisfied resulting in false positives? If not all forced executed branches are realizable, how did the authors address the potential false positives resulted by this?

For section 4.2, I suggest that the authors augment the numbers they report for the pre-analysis step with percentages.

In the last paragraph of section 4.2, the authors report that their tool identified 49 new exploitable gadgets and 378 packages that were potentially exploitable. I am a bit confused how the 378 packages relate to Table 1. Are these packages potentially exploitable due to an uncertainty about the underlying sinks? I suggest that the authors review this section (4.2) and provide a breakdown that is easier to follow.

### Update based on author(s) responses:
I appreciate the author(s) effort to respond to my questions. Overall I did not have any major concerns about the technical quality of the paper, and given the discussion about the inherent differences of Silent Spring and Dasty (e.g., operating based on limited Node.js APIs vs NPM packages) I updated my verdict.

###### [1] Shcherbakov, M., Balliu, M., & Staicu, C. A. (2023). Silent spring: Prototype pollution leads to remote code execution in Node. js. In USENIX Security Symposium 2023.

**Questions:**

Repeating my questions from the review:

- How would you handle the case where the type information cannot be extracted?
- Is writing a test as an input for the dynamic analysis a trivial task or does it require in depth knowledge of the code structure in the target npm packages? Would Dasty have identified more gadgets if the authors had access to better test cases compared to the existing unit tests that were extracted from the source code of packages under analysis?
- Would forced execution explore branches where their conditions cannot feasibly be satisfied resulting in false positives? If not all forced executed branches are realizable, how did the authors address the potential false positives resulted by this?
- For section 4.2, I suggest that the authors augment the numbers they report for the pre-analysis step with percentages.
- In the last paragraph of section 4.2, the authors report that their tool identified 49 new exploitable gadgets and 378 packages that were potentially exploitable. I am a bit confused how the 378 packages relate to Table 1. Are these packages potentially exploitable due to an uncertainty about the underlying sinks?

**Reviewer Confidence:**

3: The reviewer is confident but not certain that the evaluation is correct

**Scope:**

4: The work is relevant to the Web and to the track, and is of broad interest to the community

---

### Official Review · Reviewer_bwPE · 2023-11-22

**Novelty:** 6
**Technical Quality:** 5

**Review:**

Overall, I really enjoyed this paper. The authors implemented a dynamic taint analysis system (Dasty) for finding gadgets in node.js that can be used for prototype-pollution attacks. In their evaluation they were able to find 49 gadgets in popular real-world NPM packages. They then used the tool Silent Spring [38] to build an end-to-end exploit using Dasty for the Kibana app.

__Pros:__

1. Reasonably good results demonstrated in the evaluation for detecting usable gadgets in real-world packages, which they found 49 new gadgets.

2. They extended existing work Augur[6] in order to complete their  comparison to related work. The authors then demonstrated that Dasty outperforms Augur in terms of runtime overhead and in terms of effectives of finding gadgets.

3. The authors demonstrated the capability of their system by developing an end-to-end exploit for the popular software Kibana.

__Cons:__

1. The section _Effectiveness: Dasty vs. Augur_ is quite confusing to read and does do a good job of summarizing the results between Augur and Dasty. For example, what is meant by the "list of newly verified gadgets"? Is this the gadgets that were found in Table 1? If so, it needs to be more specific so readers will know exactly what gadgets were used in the experiments. Additionally, from my interpretation of the results, it suggests that Augur could only detect the gadgets in 3 packages, but there is no discussion into why this number is so low compared to Dasty. Given that this is the only comparison between Dasty and an existing system, the results need to be discussed more. This is necessary so that readers can fully understand the technical contributions that were made compared to existing work.

2. The paper mentions that existing static-based tools [38] have been developed for detecting gadgets in node.js, but it is unclear why this tool was not used for comparison with the related work. If this  was the case, it would have been much more clear exactly what the advantages of their approach would have been. However, this comparison is not in the paper.

3. Dasty cannot be used by itself to find a vulnerability, this in itself is fine. However, there is very litte evaluation in the paper that tries to combine using Dasty and existing tools for detecting a prototype pollution vulnerability to build an end-to-end experiment. They do provide an example in Section 4.4 (The example itself is good), but nothing empirical that would allow the reviewer/reader to understand how well Dasty would synergize with existing systems.


__Response Discussion:__ I have read the rebuttal, and overall my perspective of the paper is the same, which is that I consider this a strong paper.

**Questions:**

1. Do you all intend to open-source Dasty?

**Ethics Review Description:**

There are no ethical issues with this paper.

**Reviewer Confidence:**

3: The reviewer is confident but not certain that the evaluation is correct

**Scope:**

4: The work is relevant to the Web and to the track, and is of broad interest to the community

---

### Official Review · Reviewer_p2RG · 2023-11-24

**Novelty:** 4
**Technical Quality:** 5

**Review:**

# Strengths
- The authors implemented a compact dynamic taint analysis for detecting prototype pollution gadgets. To avoid unexpected exceptions and crashes caused by injected values in polluted property reads, Dasty used a taint proxy that can infer the expected values during the execution.

- The evaluation demonstrates Dasty's ability to identify gadgets in JavaScript code. The authors also manually validated the detected gadgets and successfully constructed end-to-end exploits.

- The authors plan to open-source their dynamic taint analysis tool.


# Weaknesses

- The authors did not discuss that Silent-spring can also detect gadgets (the second step of exploiting a prototype pollution) but simply treated it as a tool to detect polluted prototypes (the first step). The evaluation also lacks a comparison with Silent-spring in terms of their effectiveness in detecting prototype pollution gadgets.

- The authors did not qualitatively discuss the false positives and false negatives of Dasty in detecting gadgets. For example, the authors implemented force execution but did not mention if there were false positives caused by it.

- The security benefit of the techniques does not look strong as the authors presented only a high-severity CVE ID they manually discovered.


# Other comments

The paper studies an interesting problem: detecting prototype pollution gadgets in JavaScript code. The research goals are well motivated and most techniques are clearly described.

However, a critical concern is that the authors did not compare Dasty with anther work, Silent-spring, which has the same goal of detecting prototype pollution gadgets.

Additionally, the paper lacks a discussion on false positives and false negatives in its detection results.

Some claims made in this paper are inaccurate and misleading, e.g., "the first semi-automated pipeline to ...", and "For the first time, our results ...".

Important techniques such as taint propagation are not discussed in the main text but are found in the Appendix. If the authors need more space, Section 4.2 could be shorter or moved to the appendix, since performance might not be a top priority given that Dasty is not designed for runtime detection.

The paper defines sinks as all Node.js API calls, which are not all security relevant. As a result, the number of discovered gadgets could be exaggerated.

The technical detail on the proxy-based tainting technique is unclear. How is the taint proxy implemented? Is it a wrapper object that replaces the original object by performing the AST instrumentation? How and where is the taint mark injected? What is the taint mark? How do you do taint propagation in cases like internal API calls, e.g., a tainted object is passed to an JS API that returns another object?

There are some writing issues, e.t., "L732: dynamic taint analysis gadgets in Node.js".

# Post-rebuttal update

Dear authors, thanks for your responses, which have addressed most of my concerns!

Please do include the clarifications in your final version.

**Questions:**

- The authors should add a detailed comparison of Silent-spring and Dasty. Silent-spring uses static taint analysis to detect prototype pollution gadgets and Dasty uses dynamic taint analysis. It is not clear what the advantages of the two tools are, particularly in terms of their different designs and evaluation results.

- Another concern is the false positives and false negatives in Dasty's dynamic taint analysis techniques. The authors discussed a few false positive cases in Lines 635-636 but not summarized the reasons in a systematical way. For example, are there any FP cases caused by force execution? The authors might also evaluate Dasty on known gadgets (for example, show whether Dasty can detect all gadgets found by Silent-spring) to evaluate the false negatives.

**Reviewer Confidence:**

4: The reviewer is certain that the evaluation is correct and very familiar with the relevant literature

**Scope:**

4: The work is relevant to the Web and to the track, and is of broad interest to the community

---

### Decision · Program_Chairs · 2024-01-22

**Decision:**

Accept (Oral)

**Comment:**

The reviewers agreed that this paper makes an important contribution and is of broad interest to the web security community. Though some criticisms were raised in the original reviews, the discussion phase allowed the authors to properly answer the main criticisms raised by the reviewers. The authors are encouraged to take the reviewers' comments into due account when preparing the final version of their paper.

 ---